# Neuroprotective Effects of Resveratrol by Modifying Cholesterol Metabolism and Aβ Processing in SAMP8 Mice

**DOI:** 10.3390/ijms23147580

**Published:** 2022-07-08

**Authors:** Alejandro Sánchez-Melgar, Pedro J. Izquierdo-Ramírez, Christian Griñán-Ferré, Mercè Pallàs, Mairena Martín, José Luis Albasanz

**Affiliations:** 1Department of Inorganic and Organic Chemistry and Biochemistry, Faculty of Chemical and Technological Sciences, School of Medicine of Ciudad Real, Regional Center of Biomedical Research (CRIB), University of Castilla-La Mancha (UCLM), 13071 Ciudad Real, Spain; alejandro.sanchez@uclm.es (A.S.-M.); pedrojoseizquierdoramirez@gmail.com (P.J.I.-R.); mairena.martin@uclm.es (M.M.); 2Department of Pharmacology and Therapeutic Chemistry, Faculty of Pharmacy and Food Sciences, Institute of Neuroscience, University of Barcelona, 08028 Barcelona, Spain; christian.grinan@ub.edu (C.G.-F.); pallas@ub.edu (M.P.)

**Keywords:** Alzheimer’s disease, antioxidants, cell signaling, receptors/seven transmembrane domain, lipoproteins

## Abstract

Cholesterol metabolism seems dysregulated and linked to amyloid-β (Aβ) formation in neurodegeneration, but the underlying mechanisms are poorly known. Resveratrol (RSV) is a polyphenol with antioxidant activity and neuroprotective properties. Here, we analyzed the effect of age and RSV supplementation on cholesterol metabolism in the brain and blood serum, and its potential link to Aβ processing, in SAMP8 mice—an animal model of aging and Alzheimer’s disease. In the brain, our results revealed an age-related increase in ApoE and unesterified cholesterol in the plasma membrane whereas LDL receptor, HMG-CoA reductase, HMG-CoA-C1 synthase, and ABCA1 transporter remained unaltered. Furthermore, BACE-1 and APP gene expression was decreased. This dysregulation could be involved in the amyloidogenic processing pathway of APP towards Aβ formation. In turn, RSV exhibited an age-dependent effect. While levels of unesterified cholesterol in the plasma membrane were not affected by RSV, several participants in cholesterol uptake, release, and de novo synthesis differed, depending on age. Thus, RSV supplementation exhibited a different neuroprotective effect acting on Aβ processing or cholesterol metabolism in the brain at earlier or later ages, respectively. In blood serum, HDL lipoprotein and free cholesterol were increased by age, whereas VLDL and LDL lipoproteins remained unaltered. Again, the protective effect of RSV by decreasing the LDL or increasing the HDL levels also seems to depend on the intervention’s moment. In conclusion, age is a prominent factor for cholesterol metabolism dysregulation in the brain of SAMP8 mice and influences the protective effects of RSV through cholesterol metabolism and Aβ processing.

## 1. Introduction

Cholesterol, a lipidic compound present in all mammalian cells, is essential for a variety of biological functions, ranging from the precursor of steroid hormones to bile salts production. Cholesterol also takes part in cellular structures such as the plasma membrane, playing an important role by regulating membrane fluidity or by participating in several signalling pathways steps (i.e., release and re-uptake of neurotransmitters) through the interaction with sphingolipids and membrane-associated proteins [1]. Homeostasis of cholesterol is regulated by several mechanisms, namely, cellular uptake, efflux transport, endogenous synthesis, and esterification, in which many participants, such as enzymes, specific transporters, and carriers, are involved [2].

Peripheral cholesterol is mostly synthesized in the liver, whereas in the brain, cholesterol synthesis and regulation appear to be endogenous being glial cells the main producers of cholesterol as neuronal synthesis remains at a very low rate [3]. The brain is a highly cholesterol-enriched organ that contains ca. 23% of the total cholesterol in the body. This may reflect the biological significance of this lipid in the central nervous system (CNS). Moreover, transport and regulation of cholesterol metabolism within the CNS seem to be different from that in peripheral tissues since the blood–brain barrier (BBB) does not permit the entry of cholesterol. In line with this, apolipoprotein E (ApoE) has been described as the most important cholesterol carrier in the CNS rather than LDL, as occurs in peripheral blood. Besides, the ApoE ε4 allele is considered the strongest genetic risk factor for late-onset Alzheimer’s disease (AD) in humans [4], but the underlying mechanism of how ApoE influences the development of AD is still under investigation.

It is well known that a balanced content of cholesterol is crucial for maintaining cell physiology and its dysregulation may be involved in the pathogenesis of metabolic disorders, cancer, and cardiovascular and neurodegenerative diseases [5,6,7,8,9]. Accordingly, impaired homeostasis of cholesterol in the brain has been described in neurodegenerative diseases such as AD [10]. One of the most distinctive hallmarks of AD pathology is the presence of extracellular deposits of amyloid-β (Aβ), leading to neuronal damage and death [11]. Although still controversial, the idea of cholesterol as a key regulator of the amyloidogenic pathway of Amyloid Precursor Protein (APP) and Aβ formation is gaining the attention of research since statins, a drug widely used to lower peripheral cholesterol levels by inhibiting de novo synthesis, have been reported to lower the prevalence of AD for subjects taking this drug [12]. However, the precise molecular mechanism by which the homeostasis of cholesterol may influence Aβ processing is poorly understood.

The senescence-accelerated mouse (SAM)—a murine model of accelerated senescence—was successfully established by Takeda et al. [13]. SAM consists of the senescence-accelerated prone mouse (SAM-P) and senescence-accelerated resistant mouse (SAM-R), the latter of which shows normal aging characteristics. It has been demonstrated that the SAMP8 strain exhibit age-related deterioration of learning ability compared with the SAMR1 control [14]. Therefore, the SAMP8 mouse is not only a spontaneous animal model of accelerated aging but also a model of AD pathology with different features, such as memory impairment, Aβ accumulation in the hippocampus, astrogliosis, *tau* hyperphosphorylation, oxidative stress, neuroinflammation, and BBB dysfunction, among others [15,16,17].

Resveratrol (RSV), a natural polyphenol with antioxidant activity and produced by plants and present in berries, grapes, and peanuts, among others, has shown proven beneficial effects for health, such as antitumor, antioxidant, antiviral, and neuroprotective effects in AD [18,19]. It has been demonstrated that this antioxidant prevents Aβ-induced [20] as well as high-fat diet-induced cognitive decline [21], reduces LDL-oxidized levels [22], and improves lipid-lowering efficacy in hepatic cells [23]. Nevertheless, RSV triggers a plethora of signalling pathways, and the precise mode of action by which this phytochemical provides neuroprotection is still to be clarified. The effects and mechanisms of resveratrol on aging and age-related diseases, including neurodegenerative disorders, have been reviewed elsewhere [24,25]. Our previous work revealed that RSV supplementation in SAMP8 mice was able to modulate adenosine metabolism [26] and to modulate and reverse the age-related effect on adenosine-mediated signalling [27], and that metabotropic glutamate receptors are differently modulated by RSV, depending on age [28]. Moreover, we have described that a high-fat diet (HFD) and RSV diet modulate adenosinergic signalling in SAMP8 mice by increasing adenosine A_2_ receptors and 5′-NT activity [29]. Interestingly, we found evidence of the presence in serum and exosomes of adenosine A_1_, A_2A_, and mGlu5 receptors, and a potential association between brain and serum receptors levels, which were modulated by aging and resveratrol [30]

Therefore, the present work aimed to investigate whether cholesterol metabolism was altered by age and RSV supplementation in the brain and blood serum of SAMP8 mice—an animal model of aging and Alzheimer’s disease—and their potential link with the amyloidogenic pathway of APP.

## 2. Results

### 2.1. Levels of Free Cholesterol in the Brain of SAMP8 Mice

Given that cholesterol homeostasis in the brain seems to be defective in neurodegenerative disorders [31], we quantified the free cholesterol (unesterified cholesterol) levels in the brain of SAMP8 mice. As Figure 1 shows, a higher content of free cholesterol was detected in older animals, suggesting a dysregulation of cholesterol located into the plasma membrane of these animals. A two-way ANOVA was performed to analyse the effect of age and resveratrol (RSV) supplementation on free cholesterol levels. This analysis revealed that there was not a statistically significant interaction between the effects of age and RSV (F_(1, 14)_ = 0.34, *p* = 0.5715). RSV supplementation did not cause significant change in free cholesterol levels when compared with their corresponding age-matched control group (F_(1, 14)_ = 0.03, *p* = 0.8734). However, age did have a statistically significant effect on free cholesterol level (F_(1, 14)_ = 21.84, *p* = 0.0004), with a significant higher cholesterol level in control (*p* = 0.0067) and RSV-treated (*p* = 0.0165) animals.

### 2.2. Cholesterol Uptake and Release in the Brain of SAMP8 Mice

We next analysed the LDL receptor (LDL-R) level to verify whether the increase in free cholesterol associated with age previously shown was due to a higher cholesterol uptake through LDL-R. As it can be observed in Figure 2A, the precursor form of this receptor was not affected by age (F_(1, 10)_ = 0.73, *p* = 0.4116) and remained unaltered by RSV supplementation (F_(1, 10)_ = 1.88, *p* = 0.2000). Regarding the mature form of LDL-R (Figure 2B), which can be considered the active form, there was a statistically significant interaction between the effects of age and RSV (F_(1, 10)_ = 24.69, *p* = 0.0006). No age-related changes were detected in this receptor form in control animals (*p* = 0.2472). However, a strong reduction in the density of mature LDL-R is detected in 5 mo RSV-treated mice as compared to their corresponding control (*p* = 0.0004), suggesting that RSV could reduce the cholesterol uptake. However, RSV did not modulate the active form of LDL-R in 7 mo mice (*p* = 0.4168), which present significantly higher levels (*p* = 0.0009) than RSV-treated 5 mo mice. Since the age-related increase in free cholesterol levels in the brain plasma membrane could not be explained through the alteration of cholesterol uptake, we then analysed the expression of ATP-binding cassette protein A1 (ABCA1) transporter, which mediates cholesterol release from cells. Figure 2C shows that the ABCA1 protein levels were not significantly modulated in any experimental group, suggesting that cholesterol export is not a contributing factor to the modulation of plasma membrane cholesterol levels.

### 2.3. Levels of Cholesterol-Carrier ApoE in the Brain of SAMP8 Mice

To delve into the molecular mechanisms by which cholesterol metabolism is modulated from 5 to 7 months of age, we quantified the ApoE carrier levels. There was a statistically significant interaction between the effects of age and RSV (F_(1, 10)_ = 15.35, *p* = 0.0029). A discrete but significant (*p* = 0.0200) higher immunoreaction was found in the plasma membrane of older mice as compared to 5 mo mice (Figure 3A). A similar result (*p* = 0.0189) was also detected in the cytosolic fraction (Figure 3B). On the other hand, RSV treatment caused a significant decrease (*p* = 0.0077) in the ApoE level only in the plasma membrane of 7 mo mice (Figure 3A). Interestingly, a negative correlation was found between the precursor or mature forms of LDL-R and ApoE levels (Figure 3C).

### 2.4. Endogenous Synthesis of Cholesterol in the Brain of SAMP8 Mice

Next, we analysed the endogenous synthesis of cholesterol by quantifying the protein density of two key enzymes, 3-hydroxy-3-methylglutaryl-coenzyme A reductase (HMG-CoA-R), which catalyses a rate-limiting step in the synthesis of cholesterol, and the cytosolic hydroxymethylglutaryl-CoA synthase (HMG-CoA-C1). There was a statistically significant interaction between the effects of age and RSV (F_(1, 9)_ = 12.52, *p* = 0.0063) on the levels of HMG-CoA-R. No changes associated with age were observed in this enzyme in control mice (Figure 4A). In turn, a significant (*p* = 0.0097) increase was detected in 5 mo RSV-treated mice when compared to their corresponding control group, suggesting that a higher rate of de novo synthesis of cholesterol was induced by RSV. However, no significant (*p* = 0.4440) reduction in HMG-CoA-R levels was observed in 7 mo RSV-treated mice. Concerning HMG-CoA-C1 synthase levels, there was also a statistically significant interaction between the effects of age and RSV (F_(1, 10)_ = 35.06, *p* = 0.0001). The profile of HMG-CoA-C1 synthase levels (Figure 4B) was very similar to that of HMG-CoA-R. Thus, no changes were observed between 5 mo and 7 mo control animals (*p* = 0.2062), and RSV also significantly reduced the HMG-CoA-C1 synthase levels in 5 mo (*p* = 0.0021) and 7 mo (*p* = 0.0066) RSV-treated mice. These results suggest that RSV treatment causes an age-dependent modulation of these two key enzymes in the de novo synthesis of cholesterol.

### 2.5. Quantification of BACE-1 and APP Levels in the Brain from SAMP8 Mice

It is now well established that cholesterol may modulate APP processing, leading to increased production of amyloid-β (Aβ) [32]. Therefore, the potential link between the dysregulation of cholesterol metabolism and changes in amyloidogenic processing of APP was analysed in SAMP8 mice by real-time PCR to quantify APP and BACE-1 (i.e., the Aβ-generating enzyme) mRNA expression levels. There was a statistically significant interaction between the effects of age and RSV on the levels of APP mRNA (F_(1, 21)_ = 15.50, *p* = 0.0008) and the levels of BACE-1 mRNA (F_(1, 22)_ = 7.793, *p* = 0.0106). A remarkable downregulation of both APP (*p* < 0.0001) and BACE-1 (*p* < 0.0001) gene expression was associated with age in control mice (Figure 5A,B, respectively). In turn, RSV supplementation caused a different modulation of mRNA levels, depending on age. Thus, APP mRNA levels were significantly (*p* = 0.0004) reduced in 5 mo RSV-treated mice, while they remained unaltered in 7 mo RSV-treated mice (Figure 5A). This age-dependent regulation was also detected in the gene expression of BACE-1. No changes were detected in 5 mo mice while increased (*p* = 0.0443) levels of BACE-1 mRNA were observed in 7 mo RSV-treated mice. However, the density of BACE-1 protein detected by Western blotting remained unaltered in all conditions assayed (Figure 5C).

### 2.6. Levels of Free Cholesterol and Lipoproteins in Blood Serum from SAMP8 Mice

Next, we quantified the level of free cholesterol and cholesterol-related lipoproteins in peripheral blood serum. Analysis of free cholesterol levels revealed a trend to increase by age and even more after RSV supplementation (Figure 6A), showing similar results to those obtained in the brain (Figure 1). Two-way ANOVA analysis revealed that there was no statistically significant interaction between the effects of age and RSV (F_(1, 15)_ = 0.2457, *p* = 0.6273) on cholesterol levels. Although not significant (F_(1, 15)_ = 4.161, *p* = 0.0594), RSV supplementation seems to increase free cholesterol levels when compared with their corresponding age-matched control group. However, age did have a statistically significant effect (F_(1, 15)_ = 7.390, *p* = 0.0159) on free cholesterol level. Moreover, significantly (*p* = 0.0006) increased HDL levels were found in the 7 mo as compared to 5 mo control group, revealing an age-related increase in this lipoprotein, whereas RSV treatment only evoked a significant (*p* = 0.0260) increase in HDL levels in 5 mo mice (Figure 6B). There was a statistically significant interaction between the effects of age and RSV on HDL levels (F_(1, 14)_ = 9.862, *p* = 0.0072). Regarding VLDL-LDL levels, no significant changes were observed, neither with age nor RSV treatment, but a tendency toward decreased value was observed in 7 mo RSV-treated mice (Figure 6C). Thus, the relative proportion of free or lipoprotein-associated cholesterol to the global amount of cholesterol in blood serum was modified by age and resveratrol treatment (Figure 6D). Interestingly, a negative correlation between free cholesterol and HDL levels was found in control mice that turned into a positive correlation in RSV-treated mice (Figure 6E). This correlation was absent between free cholesterol and VLDL-LDL levels (Figure 6F).

## 3. Discussion

Results presented herein indicate that cholesterol metabolism and Aβ processing change with age in SAMP8 mice. Interestingly, RSV treatment had a different effect on both biological processes, depending on age. Thus, in 5-month-old SAMP8 mice, RSV supplementation resulted in increased HMG-CoA reductase and synthase protein levels, decreased LDL-R protein without altering ApoE or ABCA1 levels, reduced APP and BACE-1 gene expression, and increased serum HDL levels. In turn, RSV supplementation evoked opposite effects in 7-month-old mice; that is, it reduced HMG-CoA reductase and synthase protein levels, increased LDL-R, reduced ApoE protein, increased ABCA1 protein, increased APP and BACE-1 gene expression, and reduced serum LDL levels. Therefore, RSV could promote a neuroprotective effect by a different mechanism, depending on age (Figure 1).

As stated in the Introduction section, a SAMP8 mouse is a spontaneous animal model of accelerated aging with different features that converts SAMP8 mice into a good model of AD pathology [15,16,17]. These hallmarks seem to increase with aging, allowing us to consider this mouse model a good candidate for studying the earliest changes associated with AD pathogenesis. The lifespan for SAMP8 is around 10 months of age [33]. According to the half lifespan of a common mice strain and the maturational rates mouse vs. human, 2 mo represents a young human, and 4 mo a middle-aged individual, when our RSV treatment started. We evaluated SAMP8 mice at 5 (middle-aged, 38–47 years) and 7 months (old individual, 56–69 years) [34].

Cholesterol displays a large variety of functions in mammalian cells and it is particularly essential in the brain, in which the synaptic process can be influenced by cholesterol content located into the plasma membrane, where it not only regulates its fluidity but also modulates neurotransmission [35]. Thus, an unbalanced cholesterol content, increases as well as decreases, may result in synaptic failure and memory impairment [36,37]. In agreement, defects in brain cholesterol homeostasis can be linked to the prevalence of neurodegenerative diseases such AD, Parkinson’s disease, Niemann–Pick disease type C, and Huntington’s disease, among others [31,38,39,40]. Particularly, early studies in AD pathology described elevated levels of cholesterol in the brain of AD patients, which agrees with the epidemiological and observational studies in which statins, a cholesterol-lowering drug, seem to exhibit a positive effect against AD incidence [41,42,43,44]. Conversely, other works found a reduced content of cholesterol in several cortical areas of the human brain [45], but also in the hippocampus of SAMP8 mice [37]. Additionally, when total cholesterol was measured by enzymatic methods in the entire brain of AD patients, it was showed that there was a significant increase of this lipid as compared to the control subjects [46]. Our data revealed an age-dependent increase in free cholesterol (unesterified cholesterol) content in the plasma membrane isolated from the brain of SAMP8 mice, suggesting a dysregulation in cholesterol metabolism within the CNS, including biosynthesis, degradation into more hydrosoluble oxysterols, and cholesterol export. Elevated levels of brain cholesterol in the plasma membrane could increase membrane rigidity, which might affect the neurotransmission through a reduction of membrane-associated proteins movement (i.e., neurotransmitters receptors) from and to the synapsis. Notably, we previously found a significant age-dependent reduction in adenosine [27] and metabotropic glutamate receptors [28] together with their corresponding endogenous ligand [26] in SAMP8 mice. In addition, cholesterol modulates adenosine A_2A_ receptors expressed on the cell surface of in vitro glial cells [35]. Therefore, the alteration in GPCRs signalling pathways we observed previously in SAMP8 mice [27,28] could be related to cholesterol dysregulation. Supporting this, it has been reported a strong reduction in A_1_ receptors in the brain induced by a cholesterol-enriched diet in rabbits [47]. Nevertheless, our analysis cannot differentiate whether this increased level of cholesterol is homogeneously distributed throughout the plasma membrane or, on the contrary, it is concentrated in membrane microdomains taking part in structures enriched in cholesterol as lipid rafts, in which an altered lipidome can be observed in the context of neurodegenerative diseases [48].

The detected age-related increase in cholesterol content in the plasma membrane could be a consequence of the decrease in the ABCA1 pathway and HDL biophysical and biochemical changes leading to reduced HDL-mediated cholesterol efflux, as detected in the elderly in humans [49]. In the present work, no significant age-related changes were observed in HMG-CoA reductase and HMG-CoA synthase levels (i.e., cholesterol biosynthesis), LDL-R levels (i.e., cholesterol uptake), and ABCA1 transporter levels (i.e., cholesterol release); however, the discrete but significant increment in the major cholesterol carrier in the CNS, ApoE—induced by age—could explain, at least partially, the age-related increase in cholesterol content in the plasma membrane. ApoE allele E4 is a prominent risk factor for late-onset AD pathology [50].

In the hippocampus of SAMP8 mice, levels of Aβ increase about 100% between the ages of 4 and 12 months, with a significant increase even between 4 and 8 months [51]; however, APP and its mRNA increase at 12 months but not between 4 and 8 months [52] and between 2 and 9 months [53]. Our results in the whole brain revealed a decrease in APP gene expression from 5- to 7-month-old mice. Moreover, we did not find age-related changes in BACE-1 protein despite the reduction in BACE-1 mRNA expression. Similarly, a reduced BACE-1 mRNA expression has been also detected from 2 to 9 months in control SAMP8 mice, while BACE-1 protein levels were increased [53]. Although we cannot rule out other mechanisms, perhaps the reduced levels of mRNA could be the consequence of a compensatory mechanism to reduce Aβ formation. In line with this, although β-secretase activity has been reported to be increased age-dependently (2, 6, 12 mo) in the cerebral cortex of SAMP8, BACE1 mRNA expression did not change, while a tendency toward decreased BACE1 mRNA expression was found in the hippocampus [54]. It seems that there is not a direct correlation between mRNA expression and activity of BACE1 of SAMP8 with age. In fact, the enzymatic activity is not always correlating to the mRNA or protein expression. Thus, BACE1 activity and Aβ levels increased significantly with age in mouse, monkey, and human brain, but BACE1 protein levels were unchanged with age [55]. In agreement, increased BACE activity but not mRNA [56,57] or protein [58] expression changes were observed in the brain of Tg2576 with age.

Apart from the age-dependent increase in β-secretase activity [54], another contributing factor to the increased Aβ levels detected [51] could be cholesterol itself. Several authors have suggested that cholesterol and ApoE may influence the amyloidogenic pathway of APP, leading to Aβ accumulation and senile plaque formation [59]. Besides, Aβ also seems to alter lipid homeostasis, including cholesterol, suggesting a bidirectional link between these two compounds [32,60,61]. APP processing is very sensitive to a wide variety of lipid stimuli and all mechanistic steps in amyloidogenic/non-amyloidogenic processing of APP are affected by cholesterol (reviewed in [32]). APP has a cholesterol-binding site [62], suggesting that this lipid might be involved in APP proteolysis. It has been reported that cholesterol is a key regulator of Aβ accumulation and that a higher content of cholesterol was able to balance APP proteolysis towards the amyloidogenic pathway without affecting the key enzymes and β- and γ-secretase activities [59]. This could be consistent with our results, revealing not age-related changes in BACE-1 protein, despite the reduction in BACE-1 mRNA expression, but a higher cholesterol content that correlated well with also increased ApoE levels.

Among the multiple neuroprotective effects attributed to RSV [63], it has been shown to prevent memory loss and to reduce the Aβ burden by activating its cleavage and reducing its formation [64,65]. However, the mechanism involved in this neuroprotection is still unclear. Our study in SAMP8 mice revealed a complex scenario regarding the neuroprotective actions of RSV on cholesterol metabolism and Aβ processing. In younger mice, cholesterol uptake via LDL-R seems to be reduced sufficiently to evoke the overactivation of de novo synthesis of cholesterol through the increased levels of HMG-CoA reductase and HMG-CoA-C1 synthase to keep the content of cholesterol unchanged in the brain, as observed, while ApoE and ABCA1 remained unchanged. Interestingly, while cholesterol homeostasis seems to be insufficiently modulated to afford neuroprotection, this could be promoted by the reduced expression of APP and BACE-1 mRNAs. In turn, in older RSV-treated mice, the increase in LDL-R could be enough to reduce the ApoE levels, which would reduce cholesterol uptake. In addition, a lower de novo synthesis of cholesterol is expected since HMG-CoA synthase and HMG-CoA reductase were less abundant, which together with an enhanced cholesterol release through the higher levels of ABCA1 protein could lead to reduced cholesterol levels in the brain. However, the cholesterol content at the brain plasma membrane was not modulated by RSV. All these effects of RSV in 7 mo mice are considered neuroprotective. Thus, since ApoE primarily binds to LDL-R and LDLR-related protein 1 (LRP1) in the brain, mediating the endocytosis and clearance of ApoE lipoproteins [66], the overexpression of LDL-R is an effective approach to significantly induce ApoE reduction and consequent alleviation of pathologic changes in AD mouse models [67,68,69]. For instance, the uptake and degradation of Aβ by astrocytes were altered by modulating levels of LDL-R in the brain [70] and a higher expression of LDL-R is convenient for the clearance of toxic substances such as Aβ proteins. The links between lipids metabolism and Alzheimer’s disease reveals apolipoprotein E as a key factor [39,40,71], and these studies demonstrate that upregulation of LDL-R can lead to a reduction in ApoE and alleviation of AD pathologies (reviewed in [72]). The high level of LDL-R might likely elevate the clearance rate of Aβ and decrease the Aβ level in the SAMP8 brain. Another RSV effect contributing to its neuroprotective role could be the enhanced presence of ABCA1 in the plasma membrane. Reduced ABCA1 expression or activity is implicated in AD and other disorders. Therefore, therapeutic approaches to boost ABCA1 activity are promising [73]. It has been reported that RSV effects on cholesterol efflux via the overexpression of ABCA1 protein are mediated through the adenosine A_2A_ receptor in THP-1 macrophages and HAEC cells [74,75]. In agreement, we have previously reported that RSV acts as a non-selective adenosine receptor agonist [76] that in the same SAMP8 mice as used in the present work can differently modulate adenosine-mediated signalling, depending on age [27]. Thus, an adenosinergic mechanism may play a role in the different RSV effects on cholesterol metabolism reported here at different ages.

BBB prevents the direct exchange of cholesterol between blood and brain, and cholesterol within the brain does not readily equilibrate with cholesterol bound to lipoproteins in the blood [39]. Peripheral cholesterol (i.e., in blood serum) was also found altered by age since a higher amount of HDL and free cholesterol were found in older mice. It has been reported that RSV exhibits an LDL-lowering effect [23,77,78]; however, our work revealed that our RSV supplementation only showed a tendency toward reduced VLDL-LDL levels in 7 mo mice. In contrast, HDL levels were significantly elevated by RSV supplementation in 5 mo but not 7 mo mice. Both effects (i.e., lowering of LDL and raising of HDL levels) can be neuroprotective [79], and reveal the importance of deciding on the start or intervention with RSV to achieve such a neuroprotective effect. In line with this, a study conducted by Crandall et al. [80] reported no significant change in the lipid profile (i.e., LDL, HDL, total cholesterol) following resveratrol supplementation in older adults (aged 72 ± 3 years). Resveratrol supplementation increased the HDL levels in type 2 diabetes mellitus patients [81,82] or in obese patients with metabolic syndrome [83]. However, resveratrol did not significantly improve HDL in obese patients with type 2 diabetes [84], suggesting the importance of the health status of individuals to be treated with this polyphenol. It seems that factors such as the type of resveratrol supplement, supplementation dosage, intervention duration, and health status of the population contribute to the contradictory findings concerning the serum lipid profile and the increase [81,82,83,85,86] or no effect [87,88,89] in HDL levels after resveratrol supplementation.

## 4. Materials and Methods

### 4.1. Animals and Resveratrol Diet

A total of 26 male SAMP8 mice (ENVIGO, Barcelona, Spain) were divided into four groups: control 5 months old (mo) (n = 5), control 7 mo (n = 9), resveratrol-supplemented (RSV) 5 mo (n = 5), and RSV-supplemented 7 mo (n = 7). Control mice received a standard diet (2018 Teklad Global 18% Protein Rodent Maintenance Diet, ENVIGO, Barcelona, Spain) while RSV-supplemented mice received the same diet supplemented with trans-resveratrol (RSV) manufactured by ENVIGO (1 g RSV/kg chow, Mega Resveratrol, Candlewood Stars, Inc., Danbury, CT, USA), starting from the weaning or 4 mo for 5 and 7 mo mice, respectively (Figure 2). All the mice had food and water ad libitum and were kept in standard conditions of temperature (22 ± 2 °C) and 12:12-h light–dark cycles (300 lux/0 lux). There were no diet intake-related differences (i.e., diet taste preference). There were no significant changes in food intake between groups. Food intake was routinely controlled and revealed that, on average, each animal (ca. 30 g body weight) eats 5 g of chow by day. Therefore, this RSV supplementation results in a daily dose of 160 mg/kg (body weight). In addition, any substantial side effects (e.g., body-weight reduction, loss of appetite) were observed during RSV supplementation. All experimental procedures involving animals were performed followed by standard ethical guidelines European Communities Council Directive 86/609/EEC and by the Institutional Animal Care and Use Committee of the University of Barcelona (670/14/8102, approved on 14 November 2014) and by Generalitat de Catalunya (10291, approved on 28 January 2018). All efforts were made to minimize the number of mice used and their suffering.

### 4.2. Blood Serum Collection

Whole blood serum samples from SAMP8 mice were collected by using 4.4 mL, 75 × 13 mm, Z-Gel tubes. Blood was allowed to clot by leaving it undisturbed at room temperature, and finally, clot was removed by centrifugation at 2000× *g* for 10 min in a refrigerated centrifuge. The supernatant was collected and stored at −80 °C.

### 4.3. Brain Extraction and Plasma Membrane Isolation

Mice were euthanized by decapitation, and the brain was quickly removed from the skull. Brain plasma membranes were isolated from SAMP8 mice as previously described [27]. Cerebellum and spinal cord were excluded from whole-brain preparations, which were homogenized in 20 volumes of isolation buffer composed of 50 mM Tris-HCl, pH 7.4, 10 mM MgCl_2_, and protease inhibitors. After homogenization (Dounce 10 × A, 10 × B), samples were centrifuged for 5 min at 1000× *g* in a Beckman JA21 centrifuge. The supernatants were centrifuged at 27,000× *g* for 20 min, and the resulting pellet was resuspended in the same isolation buffer and maintained at −80 °C until used. The Lowry method was employed to measure protein levels.

### 4.4. Total RNA Extraction and cDNA Preparation

Isolation of total RNA was performed in whole-brain homogenate (excluding spinal cord and cerebellum) using an ABI6100 Nucleic Acid PrepStation following the manufacturer’s protocol no. 4330252 (Applied Biosystems, Madrid, Spain). In brief, the homogenate was first incubated in Lysis Solution (no. 4305895), the lysate was filtered through an application-specific membrane (no. 4305673), and the purified RNA was eluted. Genomic DNA was removed from samples by DNase treatment (AbsoluteRNA Wash Solution, no. 4305545) during isolation. Total RNA samples were stored at −80 °C. The purity of RNA (A_260_/A_280_ ratio) was in the range 1.7–2.3. After RNA concentration determination from A_260_ values, the corresponding cDNA was obtained by reverse transcription using a High-Capacity cDNA kit (no. 4368813) from Applied Biosystems (Madrid, Spain).

### 4.5. Gene Expression Analysis by Real-Time PCR

Real-time RT-PCR analysis was quantitative and performed in a Prism 7500 Fast system, by using TaqMan universal PCR master mix following the producer’s protocol (Applied Biosystems). TaqMan probes and primers for BACE-1 (Mm00478664_m1), APP (Mm01344172_m1), and β-actin (Mm00607939_s1) were packaged together in a 20× solution and purchased from Applied Biosystems. Gene expression was measured following the manufacturer’s protocol, as previously assayed in these mice. The thermal cycler program was 20 s at 95 °C, followed by 40 cycles of a two-step PCR (3 s at 95 °C, 30 s at 60 °C). Levels of expression were determined according to the 2^−ΔΔCt^ method. Briefly, the gene expression level was normalized to the internal control (β-actin) relative to the mean expression level of the corresponding gene in control samples (i.e., calibrator) according to the equation 2^−ΔΔCt^ = 2^−(ΔCt)sample−(ΔCt)calibrator^, where ΔCt = Ct-receptor-gene − Ct-actin-gene. These assays were performed in duplicate using different cDNAs from the animals analysed. A single mean quantity value for each analysed mRNA was obtained in each animal.

### 4.6. Immunodetection by Western Blotting Assay

Western blotting assays were carried out to detect and quantify the proteins of interest. Thirty micrograms of protein of plasma membrane fraction from each sample were mixed with loading buffer containing 0.125 M Tris (pH 6.8), 20% glycerol, 10% β-mercaptoethanol, 4% SDS, and 0.002% bromophenol blue, and heated at 50 °C for 5 min. Proteins were electrophoresed on a 10% SDS-PAGE gel using a mini-protean system (Bio-Rad, Madrid, Spain) with molecular weight standards (Bio-Rad). Protein transfer to nitrocellulose membranes was carried out in iBlot^TM^ Dry Blotting System (Invitrogen, Madrid, Spain). Nitrocellulose was washed with PBS-Tween 20, blocked with PBS containing 5% skimmed milk, and then incubated with the primary antibodies at 4 °C overnight at 1:1000 dilution for anti-LDL-R (Abcam, ab30532), anti-ApoE (Santa Cruz Biotechnology, Heidelberg, Germany, sc-13521), anti-ABCA1 (Santa Cruz Biotechnology, Heidelberg, Germany, sc-58219), anti-BACE-1 (Sigma-Aldrich, Madrid, España, WH0023621M1). Nitrocellulose membranes were also incubated with anti-beta-actin (Abcam, Cambridge, UK, ab8226) at a dilution of 1:2000, which was used as a loading control. After rinsing, the nitrocellulose membranes were incubated with the corresponding secondary antibody (Bio-Rad, GAMPO 170-6516, Madrid, Spain) at a dilution of 1:4000 in PBS containing 5% skimmed milk for 1 h at room temperature. Antigen was visualized using the ECL chemiluminescence detection kit (Amersham, Madrid, Spain) in a G:Box chamber, and specific bands were quantified by densitometry using GeneTools software (Syngene, Cambridge, UK).

### 4.7. Quantification of Free Cholesterol and Lipoproteins

Free cholesterol was quantified following the manufacturer’s indications (MAK043, Sigma-Aldrich). A total of 25 µL of plasma membrane fraction of each sample was added into 200 µL of a mixture containing Chloroform:Isopropanol:IGEPAL (7:11:0.1) for cholesterol extraction from samples. Then, samples were centrifuged at 13,000× *g* for 10 min and supernatants were transferred into new tubes. Supernatants were heated at 50 °C for 40 min to remove the organic phase from samples. A total of 20 µL of serum of each sample was used to determine the free-cholesterol present in this biological fluid. Serum lipoproteins were measured following the manufacturer’s protocol (MAK045, Sigma-Aldrich). A mixture containing sample precipitation buffer (1:1) was incubated at room temperature for 10 min and then centrifuged at 2000× *g* for 10 min. The supernatant (HDL fraction) was transferred into a new tube, and the obtained pellet was centrifuged again at 2000× *g* for 10 min to remove possible traces of HDL. A pellet was considered as the LDL/VLDL fraction. Both free cholesterol and lipoproteins were resuspended in PBS and the following steps are common in both assays. Next, samples were resuspended in the corresponding assay buffer. A total of 50 µL of resuspended samples were added into a 96-well plate with 50 µL of a reaction mix as indicated in the manufacturer’s protocol. The 96-well plate was incubated for 1 h at 37 °C and protected from light. After incubation, absorbance was measured at 570 nm with a reader plate, and data were interpolated into a standard curve.

### 4.8. Statistical and Data Analysis

Data are presented as the means ± SEM of the indicated number of samples in the figure legends. Statistical analysis was performed using two-way ANOVA and a Sidak post-test. Differences between mean values were considered statistically significant at *p* < 0.05. Correlation assessment between data values from the different experimental groups was performed by Pearson correlation analysis. Statistical and data analysis were performed with the GraphPad Prism 8.0 program (GraphPad Software, San Diego, CA, USA).

## 5. Conclusions

To sum up, our work shows that age is a prominent factor for cholesterol metabolism impairment in the SAMP8 mouse model in both brain and blood serum. Besides, this dysregulation could be involved in the amyloidogenic pathway of APP towards Aβ formation. Fortunately, RSV supplementation exhibited a different neuroprotective effect acting on Aβ processing or cholesterol metabolism in the brain at earlier or later ages, respectively. At the peripheral level, the protective effect of RSV by decreasing LDL or increasing HDL levels also seems to be dependent on the moment of intervention. Thus, the potential neuroprotective effect of RSV in SAMP8 mice could be age-dependent through a complex scenario involving opposite effects in both cholesterol and Aβ metabolism (Figure 1), confirming the link between cholesterol metabolism and Aβ processing [32,42,60].

## Data Availability

All data are contained within the manuscript.

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
