# Peer review of "Neuroprotective Effects of Resveratrol by Modifying Cholesterol Metabolism and Aβ Processing in SAMP8 Mice"

_ijms, 2022, doi:10.3390/ijms23147580_

Round 1

Reviewer 1 Report

This study describes the potential neuroprotective effect of Resveratrol in SAMP8 mice that is probably age-dependent and subject to cholesterol metabolism and Aβ processing. It is a well-designed study with interesting results and is a continuation of the research team's work. There are some points that authors must address in order to be accepted for publication.

·         The title must be changed in order to be more concise

·         Abstract Section must be checked for grammatical and syntax errors

·         Within the framework of the evaluation of the neuroprotective effect of Resveratrol and in my view, it would be better to choose SAMP8 mice at an early stage like 2 months (before they develop the characteristic pathologies) and aged like 8-9 months (they present many of the hallmarks of aging-dependent neuroinflammation). In this way, the experimental data would be more solid to verify the protective effect of RSV and possibly all the recorded differences between two age-related groups might be greater.  Thus, authors must provide an extended justification for the selected age-groups of 5 &7 months mice.

·         Authors must also mention any side-effects like body-weight reduction, loss of appetite, etc that may observe during RSV administration.

Reviewer 2 Report

Comments on manuscript No: ijms-1796003

Recommendation: Minor changes

The authors in the manuscript "Effect of resveratrol on cholesterol metabolism and amyloid 2 precursor protein processing in SAMP8 mice depends on age" successfully proved the hypothesis that the age was a prominent factor for cholesterol metabolism deregulation in the brain of SAMP8 mice and influenced the protective effects of RSV through cholesterol metabolism and Aβ processing.

The article is clearly and well written, with appropriate Introduction, methodology and discussion.

The authors are asked to explain the Section 4, Animals and resveratrol diet part, how RSV was implemented in food, how the homogenisation was performed. In addition, please explain how the calculation was performed in order to conclude that the daily dose of RSV was 160 mg/kg (body weight). Please, explain how the mentioned dose was considered as acceptable. Why was not the dose dependent experiments performed?

Reviewer 3 Report

In the manuscript Authors have described the influence of resveratrol (RSV)  application to in different age mice on cholesterol metabolism.  The aim of this work was to evaluate changes in cholesterol metabolism potentially influencing on beta-amyloid formation and fate, Such activity could be useful in therapy of neurodegenerative diseases e.g. Alzheimer’s disease (AD). This work is a continuation of previous research performed by Authors.Different  parameters were evaluated in mouse SAMP8 – spontaneous animal model of accelerated aging. The introduction based on the new literature, obtained results and their  discussion were  well described and very well illustrated (Scheme 1). In the discussion a lot of newest references were quoted.  It is an interesting and valuable scientific contribution. I recommend to publish it in IJMS after minor  changes.

To enhance manuscript value please:

In Introduction add few sentences about previously obtained results and why the present work was undertaken

-          Add very short explanation how SAMP8 were obtained, why they are used as animal model of AD (some explanation is included at page 10 in discussion, but it would be reasonable to  be more developed or placed in the introduction)

-          Where are coming from SAMP8  used in the presented work

-          Scheme 1 is very informative, but some descriptions are difficult to read. Please try to include descriptions with greater letters
